# ID-Cloak: Crafting Identity-Specific Cloaks Against Personalized Text-to-Image Generation

## Abstract

Personalized text-to-image models allow users to generate images of new concepts from several reference photos, thereby leading to critical concerns regarding civil privacy. Although several anti-personalization techniques have been developed, these methods typically assume that defenders can afford to design a privacy cloak corresponding to each specific image. However, due to extensive personal images shared online, image-specific methods are limited by real-world practical applications. To address this, we are the first to investigate the creation of identity-specific cloaks (ID-Cloak) that safeguard all images belong to a specific identity. Specifically, we first model an identity subspace that preserves personal commonalities and learns diverse contexts to capture the image distribution to be protected. Then, we craft identity-specific cloaks with the proposed novel objective that encourages the cloak to guide the model away from its normal output within the subspace. Extensive experiments show that the generated universal cloak can effectively protect the images. We believe our method, along with the proposed identity-specific cloak setting, marks a notable advance in realistic privacy protection.

## 1 Introduction

With the advent of diffusion models Ho et al. (2020); Sohl-Dickstein et al. (2015); Rombach et al. (2022), personalized text-to-image (T2I) generation Ruiz et al. (2023); Gal et al. (2023); Hu et al. (2022) has ushered a novel image generation paradigm, which enables learning new concepts to generate novel images in various contexts. However, malicious users can easily collect personal images from social media and generate offensive fabricated images. The potential privacy violations and the risk of image-based fraud have raised significant public concern Juefei-Xu et al. (2022). Developing robust algorithms to safeguard against the malicious exploitation of diffusion models is imperative for both the research community and society.

To this end, recent studies Liang et al. (2023); Van Le et al. (2023); Xue et al. (2023); Wan et al. (2024) delve into research on anti-personalization by introducing imperceptible perturbations, i.e. cloak, onto input images. Malicious users can't generate personalized images based on the protected images as the generation performance is degraded by the cloak. While these methods represent significant progress, they all follow an image-specific assumption that the cloaks generated by defenders are built upon a one-to-one correspondence with the protected images. Given the vast amounts and rapid updates of personal images accessible online, the image-specific assumption may not be realistic in practice, since each new image necessitates the reapplication of these techniques to create new privacy cloaks, rendering them highly inefficient and burdensome.

In this paper, we first investigate the robustness of current defense approaches. As shown in Fig. 1 (a), when applying cloaks generated by image-specific protection methods to other images of the same identity, the protective performance is notably weakened. This brings us to the question: ***How can we design a universal privacy cloak that can protect all images of an identity?*** This demand poses two challenges: 1) "what to protect". In the traditional "image-specific" setting, protection is applied to a predefined set of images, where the defender has full knowledge of the protection targets. However, in the "identity-specific" setting, the exact distribution of images requiring protection is unknown. Defenders are only provided with scarce samples from the target identity, which is insufficient to

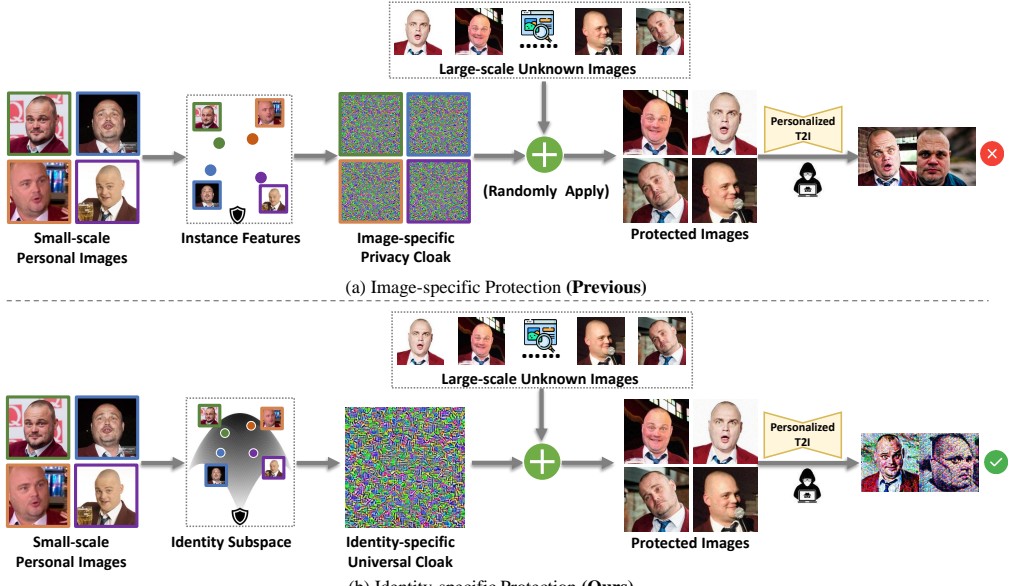

Figure 1: Comparison between image-specific and identity-specific protection. (a) Existing image-specific protection methods lose effectiveness when applied to large-scale unknown images. (b) Our identity-specific cloak exhibits effective and consistent protection.

capture the underlying image distribution that needs to be protected. 2) "how to protect". As our objective shifts from optimizing cloaks for specific "images" to broader "identities", how to optimize cloaks in an "identity-specific" manner remains an open question that requires further investigation.

To tackle these challenges, we propose ID-Cloak to generate identity-specific universal cloaks from only a few images of an individual. Specifically, we solve the "what to protect" problem by learning an identity subspace in the text embedding space. This identity subspace is intended to capture the entire distribution of the person, and implicitly covers all possible images to be protected, as illustrated in Fig. 1 (b). Specifically, we model the subspace as a Gaussian distribution, where the mean and variance are estimated from a set of anchor points in the text embedding space. These anchor points are learned via prompt learning, which capture both core identity and diverse protection contexts. To solve the "how to protect" question, we develop a novel optimization objective that encourages the cloak to steer the model away from its normal output within this modeled subspace. Qualitative and quantitative experiments demonstrate that our method can learn an identity-specific cloak from a small set of images, effectively protecting all possible images of the target identity. Our contributions are as follows:

- We introduce a novel protection paradigm against personalization misuse, shifting from image-specific to identity-specific defenses for practical usability.

- We propose ID-Cloak to craft identity-specific cloaks from a minimal set of an individual's images. It first models an identity subspace to capture the underlying protection distribution, then optimizes the cloak within this subspace using a novel optimization objective.

- Our experimental results, both qualitative and quantitative, demonstrate that our approach achieves robust protection across all images of an identity using a single universal mask.

## 2 RELATED WORK

### 2.1 PERSONALIZED T2I GENERATION

With the advance of diffusion models, current text-to-image (T2I) generation Nichol et al. (2022); Rombach et al. (2022); Wang et al. (2024b); Cui et al. (2024) has shown remarkable generalization ability. As these methods ignore the concepts that do not appear in the training set, some works study personalized text-to-image generation which aims to adapt text-to-image models to specific concepts (attributions, styles, or objects) given several reference images. Textual Inversion Gal et al.

(2023) adjusts text embeddings of a new pseudoword to describe the concept. DreamBooth Ruiz et al. (2023) fine-tunes denoising networks to connect the novel concept and a less commonly used word token. Based on that, several recent works Kumari et al. (2023); Chen et al. (2023a); Shi et al. (2024) have been proposed to enhance controllability and flexibility when processing image visual concepts. These advancements enhance the capabilities of text-to-image models, making them accessible to a wider range of users.

## 2.2 ADVERSARIAL EXAMPLES

Adversarial examples are crafted by adding imperceptible perturbations to mislead models, primarily applied in anti-classification, anti-deepfakes, and anti-facial recognition. Existing methods fall into two categories: **Image-specific adversarial examples** generate tailored perturbations per image. Szegedy (2013) pioneered this concept with LBFGS optimization, while Goodfellow et al. (2014) proposed the efficient FGSM. Subsequent works Xiao et al. (2018); Xiong et al. (2023); Chen et al. (2023b) improved naturalness via generative models. These are extended to disrupt deepfakes Ruiz et al. (2020); Wang et al. (2022a;b); Li et al. (2023) and protect facial privacy Shan et al. (2020); Yang et al. (2021); Cherepanova et al. (2021); Deb et al. (2020) from unauthorized face recognition systems. **Universal adversarial perturbations (UAPs)** apply a single perturbation to all images. Moosavi-Dezfooli et al. (2017) first revealed UAPs' existence, with Liu et al. (2023) addressing gradient vanishing via aggregation and Poursaeed et al. (2018) synthesizing UAPs via generative models. For privacy, Zhong & Deng (2022) proposed gradient-based OPOM for identity-specific protection, while Liu et al. (2025) trained generators for natural adversarial cloaks. Our work aligns with UAPs but primarily aims to protect against unauthorized personalized generation.

## 2.3 ANTI-PERSONALIZATION

The remarkable generative capability of personalized T2I generation comes with safety concerns Carlini et al. (2023); Vyas et al. (2023), particularly regarding the unauthorized exploitation of personal images. To mitigate these risks, recent studies have proposed the use of adversarial examples to counteract such safety issues. AdvDM Liang et al. (2023) pioneered a theoretical framework for crafting adversarial examples against diffusion models. Anti-DreamBooth Van Le et al. (2023) tackled anti-personalization with a bi-level protection objective and ASPL optimization, later refined by Wang et al. (2024a) via time-step selection. MetaCloak Liu et al. (2024) enhanced cloak robustness against image transformations using ensemble learning and EoT, while Xue et al. (2023) reduced computational costs via SDS loss. Li et al. (2024a) and Wan et al. (2024) addressed prompt discrepancies between protectors and attackers with encoder-based protection and prompt distribution modeling, respectively. Despite these advancements, existing methods predominantly generate image-specific cloaks, which are impractical for widespread user adoption. In contrast, our work introduces a universal cloak tailored to individual users, enabling it to be applied across all their images, significantly enhancing usability and reducing privacy risks.

## 3 METHOD

### 3.1 PROBLEM DEFINITION

We consider a scenario where a user $k$ aims to safeguard all of their current and potential future images, which are modeled as samples from the distribution $q(x)$. The user's objective is to prevent unauthorized attackers from utilizing any of their images to train personalized models for customized image generation. To achieve this, the user seeks to create a personal universal cloak $\delta$, which can be applied to any image $x \sim q(x)$. Specifically, the user applies the cloak to their images, resulting in perturbed images $x' = x + \delta$, which are then published online. The set of published images on the Internet is denoted as $X_p = \{x'_1, x'_2, \ldots, x'_i, \ldots\}$. Subsequently, an attacker may attempt to extract these images $X_p$ to train personalized models. The user's goal is to ensure that models trained on the protected images exhibit degraded generation quality. Formally, this objective is defined as:

$$\delta^* = \arg\min_{\|\delta\|_p \leq \eta} \mathcal{A}(\theta^*, k) \quad \text{s.t.} \quad \theta^* = \arg\min_{\theta} \mathbb{E}_{x' \sim X_p} \left[ \mathcal{L}_p(\theta, x') \right] \tag{1}$$

where $\theta$ denotes a pre-trained text-to-image model, $\mathcal{L}_p$ represents the personalized training objective, and $\mathcal{A}(\theta^*, k)$ is an evaluation function that assesses the quality of images generated by the personalized model $\theta^*$ with respect to the protected identity $k$. To ensure visual imperceptibility, the cloak $\delta$ is constrained within an $\ell_p$-ball of radius $\eta$.

## 3.2 ID-Cloak: Crafting Identity-Specific Cloaks

In the above problem formulation, a critical aspect is the characterization of the personal image distribution $q(x)$. Since the real distribution is unavailable, We can only describe this distribution based on the available set of personal face images $X_c \sim q(x)$ provided by the user. While a larger set of images would allow for a more accurate estimation of the distribution, obtaining a vast number of individual images is often impractical and contrary to our initial objective. Therefore, we aim to estimate $q(x)$ using $X_c = \{x_i\}_{i=1}^N$ with a limited number of images $N$.

The fundamental intuition is that a more precise approximation of the personal image distribution enhances the universality of the cloak, thereby improving its transferability across different images. Building on this, we propose ID-Cloak, a novel method for generating such identity-specific universal cloaks using a small set of an individual's images. Our approach comprises two main stages: 1) identity subspace modeling: utilizing the input few-shot images, we learn a subspace in the text embedding space which represents the individual. This subspace, together with a T2I generative model, is intended to capture the entire personal image distribution for protection. 2) universal cloak optimization: based on the modeled subspace, we develop an optimization objective to that encourages the cloak to steer the model away from its normal output within the subspace.

### 3.2.1 Modeling the Identity Subspace

We base our approach on the following assumption: Let an individual $k$ possess a protected image distribution $q(x)$ defined over the image space $\mathcal{X}$. Consider a text-to-image diffusion model with parameters $\theta$, characterized by its conditional sampling distribution $p_\theta(x|c)$, where $c \in \mathcal{C}$ denotes text conditions in the text embedding space $\mathcal{C}$. We hypothesize the existence of a latent identity subspace $Q(c)$ defined over $\mathcal{C}$ that encapsulates all text conditions semantically associated with the protected identity of individual $k$. By combining $Q(c)$ with the conditional sampling distribution $p_\theta(x|c)$, we can approximate the image distribution $q(x)$ as:

$$p_\theta(x) = \int p_\theta(x|c)Q(c)\,dc. \tag{2}$$

This formulation shifts the focus from complex image distributions to a more structured and interpretable text-based representation. Specifically, it allows us to model the protected identity's image distribution by focusing on a semantically meaningful subspace in the text embedding space.

The ideal subspace $Q(c)$ should capture both the commonalities (core identity information of the individual) and the variations (diversity of protection contexts, such as backgrounds, poses, illuminations, and expressions) to cover all potential protection scenarios. To approximate $Q(c)$, we model it as a Gaussian distribution $\hat{Q}(c)$, parameterized by the mean and variance of a set of anchor points in the text embedding space. These anchor points are learned via prompt tuning, initialized from the core identity point and optimized to associate with specific image instances, thereby incorporating diverse protection contexts. We adopt this approximation for two reasons: 1) Gaussian distribution effectively models large sample sizes, ideal for subspace representation. 2) It aligns with our ideal text embedding distribution, concentrated around the core identity point.

We begin by learning the core identity information of the individual from a few-shot set of input images $X_c = \{x_i\}_{i=1}^N$. To achieve this, we define a unique identifier $V^*$ that represents the person's identity. This identifier is combined with a sentence template to form a textual description $P$ (e.g., "a photo of $V^*$ person"). We implant the identity information into $V^*$ by optimizing the following objective, as described in Ruiz et al. (2023):

$$\min_{\theta,\tau} \mathbb{E}_{x \sim X_c, \epsilon, t} \left[ \|\epsilon - \epsilon_\theta(x_t, t, \tau(P))\|_2^2 \right], \tag{3}$$

where $\tau(\cdot)$ is a text encoder producing text embeddings. In this stage, we optimize the full model parameters $\theta$ with the text encoder $\tau$ to ensure high fidelity to the original identity, resulting in a personalized model $\theta^*$ with text encoder $\tau^*$.

---

**Algorithm 1** Learning Identity Subspace

---

**Require:** Personal images $X_c = \{x_i\}_{i=1}^N$, diffusion model $\theta$ with text encoder $\tau$, identity learning steps $C$, prompt tuning steps $M$, identity descriptor $V^*$
1: Construct textual description $P$ using $V^*$          ▷ Step 1: identity token learning
2: **for** $i = 1$ **to** $C$ **do**
3:     Sample $x \sim X_c, \epsilon \sim \mathcal{N}(\mathbf{0}, \mathbf{I}), t \in U(1, T)$
4:     $x_t = \sqrt{\alpha_t} x_0 + \sqrt{1 - \alpha_t} \epsilon$
5:     Take a gradient step on $\nabla_{\theta, \tau} \|\epsilon - \epsilon_\theta(x_t, t, \tau(P))\|_2^2$
6: **end for**
7: **Yield:** personalized model $\theta^*$, text encoder $\tau^*$
8: Initialize anchors with learned identity $\{c_i = c_{ID} = \tau^*(P)\}_{i=1}^N$ ▷ Step 2: context diversification
9: **for** $i = 1$ **to** $M$ **do**
10:     Sample $\epsilon \sim \mathcal{N}(\mathbf{0}, \mathbf{I}), t \in U(1, T)$
11:     **for** $j = 1$ **to** $N$ **do**
12:        $c_j \leftarrow c_j - \nabla_{c_j} \|\epsilon - \epsilon_{\theta^*}(x_{j,t}, t, c_j)\|_2^2$
13:     **end for**
14: **end for**
15: Construct identity subspace $\hat{Q}(c) \sim \mathcal{N}(\mu(\{c_i\}_{i=1}^N), \sigma(\{c_i\}_{i=1}^N))$
16: **Return:** Identity subspace $\hat{Q}$, personalized model $\theta^*$

---

Subsequently, to exploit the diversity inherent in the input image set, we optimize a set of soft embeddings $\{c_i\}_{i=1}^N$, where each $c_i$ is associated with a specific image $x_i$ in the input set. These embeddings are initialized with $c_{ID}$, which represents the core identity learned from the previous stage: $c_{ID} = \tau^*(P)$. Our goal is to compute $\{c_i\}_{i=1}^N$ such that each $c_i$ best describes its corresponding image $x_i$. Formally, this can be expressed as:

$$c_i = \arg\max_c p(c|x_i, c_{ID}). \tag{4}$$

However, directly maximizing this likelihood is intractable. Following prior work Chen et al. (2024); Wan et al. (2024); Zhang et al. (2025), we reformulate the problem as an expectation minimization task (refer Appendix C for details):

$$\min_{\{c_i\}_{i=1}^N} \mathbb{E}_{\epsilon, t} \sum_{i=1}^N \|\epsilon - \epsilon_{\theta^*}(x_{i,t}, t, c_i)\|_2^2. \tag{5}$$

Finally, using the obtained soft embeddings $\{c_i\}_{i=1}^N$, we approximate the identity subspace as a Gaussian distribution:

$$\hat{Q}(c) \sim \mathcal{N}\left(c; \mu(\{c_i\}_{i=1}^N), \sigma(\{c_i\}_{i=1}^N)\right), \tag{6}$$

where $\mu(\{c_i\}_{i=1}^N)$ and $\sigma(\{c_i\}_{i=1}^N)$ are the mean and standard deviation estimated in the text encoder space. Once the subspace $Q$ is established, we approximate the target distribution $q(x)$ by sampling images from the subspace using the diffusion model. This is achieved through a Monte Carlo sampling approach: $x \sim p_{\theta^*}(x|c)$, where $c \sim Q(c)$.

### 3.2.2 Optimizing Identity-Specific Cloaks

In the previous section, we approximated the real personal image distribution $q(x)$ using a personalized diffusion model $\theta^*$ parameterized by an identity subspace $Q(c)$. Our current objective is to optimize a universal cloak $\delta$ to ensure that the cloak remains adversarial across the entire personal image distribution $q(x)$. Formally, we seek for a cloak $\delta$ that maximizes the divergence between the model's output distribution under the cloaked input $p_{\theta^*}(x + \delta)$ and the personal image distribution $q(x)$. This is achieved by maximizing the cross-entropy between the two distributions:

$$\delta := \arg\max_\delta -\mathbb{E}_{q(x)} \log p_{\theta^*}(x + \delta). \tag{7}$$

This objective is closely aligned with the standard training objective of diffusion models Sohl-Dickstein et al. (2015); Ho et al. (2020), but with an adversarial intent: instead of learning to generate samples from $q(x)$, we aim to disrupt the model's ability to accurately represent the personal image

---

**Algorithm 2** Optimizing Identity-specific Universal Cloak

---

**Require:** Customized diffusion model $\theta^*$, personal subspace $\hat{Q}(c)$, training iterations $N$, noise budget $\eta$, PGD step size $\alpha$
1: **Initialize:** $\delta = 0$
2: **for** $n = 1$ **to** $N$ **do**
3:      Sample $c \sim \hat{Q}(c), t \in U(0, T)$
4:      Sample $x_t = \texttt{sample}(\theta^*, t, c)$
5:      Obtain $\hat{x}'_t = \texttt{applyCloak}(x_t, \delta)$
6:      Compute grad $g = \nabla_{\hat{x}'_t} \|\epsilon_{\theta^*}(x_t, t, c) - \epsilon_{\theta^*}(\hat{x}'_t, t, c)\|_2^2$
7:      $\delta \leftarrow \text{clip}_\delta^\eta(\delta + \alpha \cdot \text{sgn}(g))$
8: **end for**
9: **Return:** Universal cloak $\delta^*$ for identity $k$

---

distribution. Following Ho et al. (2020), we reformulate the cross-entropy objective into a tractable denoising score matching loss:

$$\max_\delta \mathbb{E}_{x \sim q(x), \epsilon, t} \left[ \|\epsilon - \epsilon_{\theta^*}(x'_t, t, c)\|_2^2 \right], \tag{8}$$

where $x'_t = \sqrt{\alpha_t}(x + \delta) + \sqrt{1 - \alpha_t}\epsilon$. However, directly optimizing this objective requires sampling from $q(x)$, which is intractable. Instead, we leverage the fact that the personalized model $\theta^*$—trained to approximate $q(x)$ via its subspace $Q(c)$—provides an accessible surrogate distribution $p_{\theta^*}(x|c)$. By sampling from $p_{\theta^*}(x|c)$ where $c \sim Q(c)$, we can effectively approximate $q(x)$. This transforms the objective into a tractable form:

$$\max_\delta \mathbb{E}_{x \sim p_{\theta^*}(x|c), c \sim Q(c), \epsilon, t} \|\epsilon_{\theta^*}(x_t, t, c) - \epsilon_{\theta^*}(x'_t, t, c)\|_2^2, \tag{9}$$

where $x_t = \sqrt{\alpha_t}x + \sqrt{1 - \alpha_t}\epsilon$. Directly optimization of Eq. (9) requires sampling from the full reverse diffusion chain $p(x_T) \prod_{t=1}^T p_\theta(x_{t-1}|x_t)$, which incurs prohibitive computational costs. Observing that while the cloak $\delta$ needs to be ultimately applied to the clean image $x$, the other terms in Eq. (9) only depend on the intermediate noisy latent $x_t$ sampled from the reverse process $T \to t$. To bypass the costly reverse process ($t \to 0$) and redundant forward passes ($0 \to t$), we propose a one-step latent cloaking strategy leveraging DDIM Song et al. (2021). Specifically, given a noisy latent $x_t$, we first estimate a clean image $\hat{x}_0$ through deterministic denoising:

$$\hat{x}_0 = \frac{x_t - (\sqrt{1 - \alpha_t})\epsilon_{\theta^*}(x_t, t, c)}{\sqrt{\alpha_t}}. \tag{10}$$

Next, the cloak $\delta$ is applied to $\hat{x}_0$, yielding the perturbed estimate $\hat{x}'_0 = \hat{x}_0 + \delta$. This perturbed estimate $\hat{x}'_0$ is then reprojected to the noisy latent space at timestep $t$ through:

$$\hat{x}'_t = \sqrt{\alpha_t}\hat{x}'_0 + \sqrt{1 - \alpha_t}\epsilon_{\theta^*}(x_t, t, c). \tag{11}$$

Finally, our target becomes:

$$\max_\delta \mathbb{E}_{x_t, c \sim Q(c), t} \left[ \|\epsilon_\theta(x_t, t, c) - \epsilon_\theta(\hat{x}'_t, t, c)\|_2^2 \right]. \tag{12}$$

By maximizing the discrepancy between noise predictions for the original and perturbed latents, the cloak is encouraged to guide the model output away from its normal behavior within the subspace across diffusion timesteps, ultimately causing the final generated images to deviate significantly from the original ones. The algorithm is outlined in Alg. (2). When updating the cloak, the Projected Gradient Descent (PGD) Madry (2017) and Stochastic Gradient Aggregation (SGA) technique Liu et al. (2023) are employed to improve gradient stability and enhance the optimization efficacy. Further details are provided in the Appendix D.

## 4 EXPERIMENT

### 4.1 EXPERIMENT SETUP

**Datasets.** We select two face datasets: CelebA-HQ Liu et al. (2015) and VGGFace2 Cao et al. (2018). For a comprehensive evaluation, we randomly select 50 identities from each dataset. For each identity,

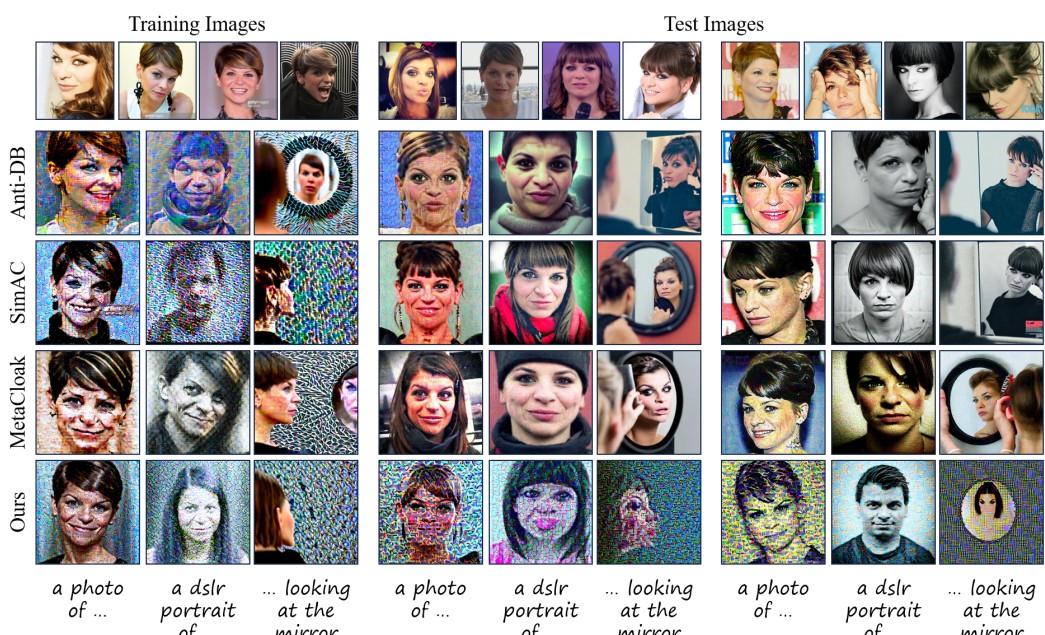

Figure 2: Qualitative results on VGGFace2 dataset. The cloaks are generated from the images of training set, then applied on the same training set and different test sets respectively. Each row represents a method, and each column represents a different test prompt.

we randomly pick 12 images and split them into two subsets: a training set and a test set. The training set is used to generate the adversarial privacy cloak, while the test set is used to evaluate the protection performance of the generated cloak. The training set and test set comprise 4 images and 8 images respectively, allowing us to thoroughly assess the effectiveness of identity-specific cloaks.

**Evaluation metrics.** Our method aims to disrupt target personalized models, causing them to generate poor-quality images of the protected identity. To evaluate the effectiveness, we design metrics that assess two key aspects of distortion: semantic-related distortion and quality-related distortion. For semantic-related distortion, our goal is to significantly alter the subject's identity in generated images to prevent misuse. We first evaluate subject detectability using the Face Detection Failure Rate (FDFR) with RetinaFace Deng et al. (2020) as the detector. If a face is detected, we then measure semantic similarity via Identity Score Matching (ISM), which computes cosine similarity between ArcFace embeddings of the generated and original faces. For quality-related distortions, we aim to degrade generated image quality. To measure this, we adopt two metrics: BRISQUE Mittal et al. (2012), widely used for image quality assessment, and SER-FIQ Terhorst et al. (2020), designed for facial image quality evaluation. To measure the defense's effectiveness, we generate 30 images using three different prompts for each trained personalized model.

**Baselines.** We conduct a comprehensive comparison of our proposed method against several state-of-the-art baselines, including Anti-DreamBooth Van Le et al. (2023), SimAC Wang et al. (2024a), and MetaCloak Liu et al. (2024). Notably, these existing methods primarily concentrate on generating image-specific privacy cloaks. To make a fair comparison, we extend these methods to generate universal cloaks by employing a gradient-averaging update strategy Moosavi-Dezfooli et al. (2016; 2017). We denote these improved variants as *Universal* methods and their original counterparts as *Image-specific* methods. For the *Image-specific* methods, we randomly transfer the cloaks learned from training images to other images in the test set, thereby constructing the final protected images for personalized fine-tuning. In contrast, for the *Universal* methods, we directly apply the cloak generated on the training set to the images in the test set, yielding the final protected images for personalized fine-tuning.

## 4.2 COMPARISON WITH BASELINE METHODS

Qualitative results are presented in Figure 2. We conduct experiments on both the training and test sets. It is evident that cloaks generated by other methods are effective only on the training set which

Table 1: Comparison with other open-sourced anti-personalization methods on VGGFace2 (left) and CelebA-HQ (right). We evaluate the performance under three different prompts during personalization. *Universal* versions are denoted with "+".

| Method | VGGFace2 | | | | CelebA-HQ | | | |
|---|---|---|---|---|---|---|---|---|
| | BRISQUE↑ | ISM↓ | FDFR↑ | SER-FIQ↓ | BRISQUE↑ | ISM↓ | FDFR↑ | SER-FIQ↓ |
| | *"a photo of sks person"* | | | | *"a photo of sks person"* | | | |
| Anti-DB | 32.132 | 0.600 | 0.004 | 0.730 | 29.160 | 0.543 | 0.002 | 0.707 |
| SimAC | 34.937 | 0.590 | 0.007 | 0.731 | 28.664 | 0.519 | 0.003 | 0.698 |
| MetaCloak | 26.616 | 0.524 | 0.008 | 0.699 | 30.074 | 0.461 | 0.007 | 0.658 |
| Anti-DB+ | 35.493 | 0.577 | 0.008 | 0.722 | 31.280 | 0.447 | 0.006 | 0.648 |
| SimAC+ | 38.164 | 0.575 | 0.010 | 0.694 | 30.409 | 0.442 | 0.007 | 0.644 |
| MetaCloak+ | 24.550 | 0.507 | 0.009 | 0.674 | 28.719 | **0.416** | 0.007 | 0.637 |
| Ours | **38.472** | **0.469** | **0.143** | **0.557** | **38.599** | 0.477 | **0.025** | **0.632** |
| | *"a dslr portrait of sks person"* | | | | *"a dslr portrait of sks person"* | | | |
| Anti-DB | 10.519 | 0.434 | 0.012 | 0.714 | 5.008 | 0.432 | 0.004 | 0.749 |
| SimAC | 10.688 | 0.444 | 0.021 | 0.708 | 4.751 | 0.432 | 0.007 | 0.747 |
| MetaCloak | 12.461 | 0.447 | 0.013 | 0.701 | 10.237 | 0.424 | 0.011 | 0.750 |
| Anti-DB+ | 11.807 | 0.414 | 0.016 | 0.706 | 6.863 | 0.427 | 0.004 | 0.747 |
| SimAC+ | 18.466 | 0.424 | 0.043 | 0.690 | 6.368 | 0.446 | 0.005 | 0.754 |
| MetaCloak+ | 11.443 | 0.449 | 0.013 | 0.706 | 9.501 | 0.412 | 0.018 | 0.746 |
| Ours | **26.143** | **0.336** | **0.228** | **0.557** | **21.304** | **0.363** | **0.055** | **0.684** |
| | *"a photo of sks person looking at the mirror"* | | | | *"a photo of sks person looking at the mirror"* | | | |
| Anti-DB | 15.427 | 0.400 | 0.064 | 0.549 | 15.189 | 0.420 | 0.055 | 0.586 |
| SimAC | 22.163 | 0.425 | 0.048 | 0.563 | 17.559 | 0.413 | 0.055 | 0.589 |
| MetaCloak | 20.221 | 0.413 | 0.060 | 0.564 | 22.009 | 0.429 | 0.053 | 0.585 |
| Anti-DB+ | 18.423 | 0.405 | 0.069 | 0.543 | 17.577 | 0.406 | 0.051 | 0.584 |
| SimAC+ | 27.421 | 0.388 | 0.073 | 0.531 | 20.483 | 0.402 | 0.053 | 0.568 |
| MetaCloak+ | 19.526 | 0.413 | 0.067 | 0.557 | 21.920 | 0.418 | 0.057 | 0.581 |
| Ours | **28.537** | **0.288** | **0.259** | **0.388** | **28.375** | **0.340** | **0.111** | **0.498** |

is the default setting in previous methods. When applied to other images of the same individual, the protective effectiveness diminishes significantly or is even completely lost. In contrast, our method consistently provides effective protection across both the training and test sets. The results with different prompts further demonstrate the robustness of our method.

The quantitative comparisons on the VGGFace2 dataset and the CelebA-HQ dataset are presented in Table 1. Our method consistently outperforms all baseline approaches across all prompts and datasets. Notably, the *universal* cloak generation variants exhibit superior performance compared to their original *image-specific* counterparts. This finding suggests that, under the setting of learning identity-specific universal cloaks, learning a single universal cloak is more effective than learning image-specific cloaks. However, our method still demonstrates a significant performance improvement over these *universal* methods. For instance, in the critical metric of FDFR, which measures whether a face can be detected in the generated image, our method, ID-Cloak, achieves an average improvement factor of **5.0** and **2.4** compared to the previous state-of-the-art methods on VGGFace2 and CelebA-HQ datasets, respectively. Even when a face is detectable, the ISM and SER-FIQ metrics indicate that our ID-Cloak generates faces with the greatest identity deviation from the original, while achieving the lowest quality for the face portion of the image. Additionally, the results on the BRISQUE metric suggest that ID-Cloak effectively degrades the overall image quality of the generated images. These results validate our method's effectiveness in creating identity-specific cloaks for robust facial privacy protection and demonstrate strong generalization across individual face images.

## 4.3 TRANSFER EXPERIMENTS

In real-world applications, attackers may employ models or personalization techniques different from those used by protectors. In this section, we conduct a series of transferability experiments. 1) model transferability: we investigate whether the privacy cloaks generated for one model can effectively protect against exploitation by other models. 2) personalization techniques transferability: we analyze whether the cloaks remain effective when attackers apply different personalization techniques.

**Model transferability.** To evaluate the transferability of the cloaks across different target models, we specifically investigate the transferability between Stable Diffusion v1.5 and Stable Diffusion v2.1. We assess the effectiveness of cloaks learned on Stable Diffusion v2.1 when applied to the personalization process on Stable Diffusion v1.5, and vice versa. The results are summarized in Table 2. Our findings demonstrate that ID-Cloak exhibits strong transferability between these two

Table 2: Transferability results across models.

| Train | Test | "a photo of sks person" | | | |
|---|---|---|---|---|---|
| | | BRISQUE↑ | ISM↓ | FDFR↑ | SER-FIQ↓ |
| v2.1 | v2.1 | 38.472 | 0.469 | 0.143 | 0.557 |
| v2.1 | v1.5 | 28.761 | 0.389 | 0.426 | 0.411 |
| v1.5 | v2.1 | 37.231 | 0.470 | 0.124 | 0.546 |

| Train | Test | "a dslr portrait of sks person" | | | |
|---|---|---|---|---|---|
| | | BRISQUE↑ | ISM↓ | FDFR↑ | SER-FIQ↓ |
| v2.1 | v2.1 | 26.143 | 0.336 | 0.228 | 0.557 |
| v2.1 | v1.5 | 11.026 | 0.375 | 0.012 | 0.634 |
| v1.5 | v2.1 | 24.139 | 0.343 | 0.176 | 0.586 |

| Train | Test | "a photo of sks person looking at the mirror" | | | |
|---|---|---|---|---|---|
| | | BRISQUE↑ | ISM↓ | FDFR↑ | SER-FIQ↓ |
| v2.1 | v2.1 | 28.537 | 0.288 | 0.259 | 0.388 |
| v2.1 | v1.5 | 26.695 | 0.343 | 0.193 | 0.488 |
| v1.5 | v2.1 | 27.326 | 0.300 | 0.240 | 0.406 |

Table 3: Transferability results across different personalization techniques.

| Method | "a photo of sks person" | | | |
|---|---|---|---|---|
| | BRISQUE↑ | ISM↓ | FDFR↑ | SER-FIQ↓ |
| DreamBooth | 38.472 | 0.469 | 0.143 | 0.557 |
| DreamBooth-LoRA | 45.990 | 0.230 | 0.487 | 0.260 |
| Textual Inversion | 59.309 | 0.273 | 0.411 | 0.560 |

| Method | "a dslr portrait of sks person" | | | |
|---|---|---|---|---|
| | BRISQUE↑ | ISM↓ | FDFR↑ | SER-FIQ↓ |
| DreamBooth | 26.143 | 0.336 | 0.228 | 0.557 |
| DreamBooth-LoRA | 31.812 | 0.189 | 0.277 | 0.452 |
| Textual Inversion | 29.947 | 0.218 | 0.149 | 0.606 |

| Method | "a photo of sks person looking at the mirror" | | | |
|---|---|---|---|---|
| | BRISQUE↑ | ISM↓ | FDFR↑ | SER-FIQ↓ |
| DreamBooth | 28.537 | 0.288 | 0.259 | 0.388 |
| DreamBooth-LoRA | 31.129 | 0.139 | 0.359 | 0.265 |
| Textual Inversion | 50.114 | 0.229 | 0.458 | 0.456 |

model versions. Across all metrics, performance remains stable in different cross-model settings, with only slight degradation. This robustness can likely be attributed to the similarity in the condensed representations of images across the models. These results demonstrate the cloaks generated by our method provide robust, broad protection across different models.

**Personalization techniques transferability.** To evaluate the robustness of the proposed method against different personalization techniques, we apply ID-Cloak to three widely adopted personalization techniques: Dreambooth Ruiz et al. (2023), Dreambooth with LoRA Hu et al. (2022) and Textual Inversion Gal et al. (2023). Dreambooth corresponds to the default configuration of our above experiments. LoRA is a widely adopted low-rank personalization method suited for low computational resources. Textual Inversion customizes concepts by optimizing a word vector rather than fine-tuning the entire model. As shown in Table 3, our method, ID-Cloak, effectively defends against both Dreambooth, LoRA and Textual Inversion, highlighting its efficacy in countering various personalization techniques.

## 4.4 ABLATION STUDY

**Effectiveness of proposed components.** To evaluate the individual contributions of ID-Cloak's components to its overall effectiveness, we conducted ablation studies on the VGGFace2 dataset. We first ablated all components and directly optimized a single cloak using the input images via the gradient-

Table 4: Ablation results of ID-Cloak.

| Sub. | Obj. | BRISQUE↑ | ISM↓ | FDFR↑ | SER-FIQ↓ |
|---|---|---|---|---|---|
| | | 22.71 | 0.468 | 0.035 | 0.649 |
| | ✓ | 30.29 | 0.364 | 0.186 | 0.506 |
| ✓ | ✓ | **31.05** | **0.364** | **0.210** | **0.501** |

averaging method Moosavi-Dezfooli et al. (2016; 2017). Next, we tested a simplified version where a single point was used to represent the individual in the text embedding space, rather than modeling a subspace. The results in Table 4 demonstrate that all simpler or alternative configurations yield inferior performance compared to our full model. Specifically, the results of the second ablation study indicate that using a single point to describe an individual's identity lacks diversity and is prone to overfitting, failing to capture the full distribution of an individual's characteristics. In contrast, modeling a subspace by incorporating the diversity of protection contexts enables coverage of a broader range of potential protection scenarios, thereby enhancing the generalizability of the cloak.

## 5 CONCLUSION

This paper introduces identity-specific cloaks (ID-cloaks), a novel privacy protection paradigm against misuse in personalized text-to-image generation. We formalize the task and propose an effective method for generating such cloaks. It first models an identity subspace in the text conditioning space to approximate the protection distribution, then optimizes universal masks utilizing a novel objective. Extensive experiments demonstrate the effectiveness of our solution, offering a scalable and practical advancement in privacy protection for generative models.

## ETHICS STATEMENT

This work adheres to the ICLR Code of Ethics. Our research involves no human subjects, crowd-sourcing, or collection of personally identifiable data. We use only publicly available datasets with proper licenses, and all third-party assets (code, data, models) are appropriately credited with their original sources and terms of use clearly respected. The proposed method is a defensive technique aimed at protecting individual identity from unauthorized personalized generation, thereby enhancing privacy and user control. While generative models can be misused, our approach mitigates such risks by preventing unauthorized exploitation of personal images. No high-risk models or sensitive scraped data are released, and thus no additional safeguards beyond standard academic practice are required.

## REPRODUCIBILITY STATEMENT

We are committed to reproducibility. All theoretical results are accompanied by clearly stated assumptions and complete proofs (provided in the main text and supplementary material). For experiments, we fully disclose model architectures, training protocols, hyperparameters, data splits, optimizer settings. The code and preprocessed datasets will be released publicly under an open-source license upon acceptance, with detailed instructions to reproduce all main results. Anonymized versions of the code and data are included in the supplementary material for review. Scripts to replicate both our method and baseline comparisons are provided.

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

## A  ADDITIONAL BACKGROUND

**Diffusion Models**. Diffusion models (Sohl-Dickstein et al., 2015; Ho et al., 2020) are a type of generative models that learns the data distribution via two opposing procedures: a forward pass and a backward pass. Given an input image $\mathbf{x}_0 \sim q(\mathbf{x})$, the forward process gradually corrupts the data over $T$ timesteps by adding Gaussian noise:

$$\mathbf{x}_t = \sqrt{\alpha_t}\mathbf{x}_0 + \sqrt{1-\alpha_t}\boldsymbol{\epsilon}, \quad \boldsymbol{\epsilon} \sim \mathcal{N}(\mathbf{0}, \mathbf{I}), \tag{13}$$

where $\{\alpha_t\}_{t=1}^T$ follows a predefined variance schedule controlling noise levels at each timestep $t \in [1, T]$. The reverse process reconstructs $\mathbf{x}_0$ from $\mathbf{x}_T$ by iteratively predicting and removing noise. A parameterized network $\boldsymbol{\epsilon}_\theta(\mathbf{x}_t, t)$ is used to estimate the noise $\boldsymbol{\epsilon}$ added at timestep $t$. The training loss is commonly defined as the $\ell_2$ distance between predicted and actual noise:

$$\mathcal{L}(\theta, \mathbf{x}_0) = \mathbb{E}_{\mathbf{x}_0, t, \boldsymbol{\epsilon} \sim \mathcal{N}(0, \mathbf{I})} \|\boldsymbol{\epsilon} - \boldsymbol{\epsilon}_\theta(\mathbf{x}_t, t)\|_2^2, \tag{14}$$

where $t$ is uniformly sampled from $\{1, \dots, T\}$.

Text-to-image diffusion models incorporates an additional conditioning signal $c$ (e.g., text prompts) into the noise prediction network:

$$\mathcal{L}(\theta, \mathbf{x}_0, c) = \mathbb{E}_{\mathbf{x}_0, t, \boldsymbol{\epsilon} \sim \mathcal{N}(0, \mathbf{I})} \|\boldsymbol{\epsilon} - \boldsymbol{\epsilon}_\theta(\mathbf{x}_t, t, c)\|_2^2. \tag{15}$$

Sampling from a diffusion model is an iterative reverse process that progressively denoises the data. Denoising Diffusion Implicit Model (DDIM) Song et al. (2021) is one of the denoising approaches with a deterministic process: Following the sampling process of DDIM, the denoising step at $t$ is formulated as:

$$\mathbf{x}_{t-1} = \sqrt{\alpha_{t-1}} \underbrace{\left( \frac{\mathbf{x}_t - \sqrt{1-\alpha_t}\boldsymbol{\epsilon}_\theta(\mathbf{x}_t, t)}{\sqrt{\alpha_t}} \right)}_{\text{Predicted } \mathbf{x}_0} + \sqrt{1-\alpha_{t-1}-\sigma_t^2}\boldsymbol{\epsilon}_\theta(\mathbf{x}_t, t) + \sigma_t\boldsymbol{\epsilon}_t, \tag{16}$$

where $\boldsymbol{\epsilon}_t \sim \mathcal{N}(\mathbf{0}, \mathbf{I})$. In our work, we utilize the denoising diffusion implicit model (DDIM) to predict the clean data point.

Recently, **Latent Diffusion Models (LDMs)** Rombach et al. (2022) have introduced a novel paradigm by operating in the latent space rather than directly in the high-dimensional data space. Specifically, the source latent variable $z_0$ is obtained by encoding a sample $x_0$ using an encoder $\mathcal{E}$, such that $z_0 = \mathcal{E}(x_0)$. This latent representation can then be reversed to reconstruct the original output through a decoder $\mathcal{D}$. By conducting the diffusion process in a lower-dimensional latent space, LDMs significantly reduce the computational burden while maintaining the quality of the generated images, making it a promising method for high-resolution image synthesis. The training of latent diffusion models involves a denoising process in the latent space, which is optimized through the following objective function:

$$\mathcal{L}_{LDM}(\theta, \mathbf{x}_0, c) = \mathbb{E}_{z_0 = \mathcal{E}(\mathbf{x}_0), \epsilon, t}[\|\boldsymbol{\epsilon} - \boldsymbol{\epsilon}_\theta(z_t, t, c)\|_2^2] \tag{17}$$

## B  IMPLEMENTATION DETAILS

For a fair comparison, we set the same noise budget for all methods, with $\eta = 16/255$, and the optimization steps and step sizes are aligned with the optimal settings specified in each baseline. The default base model used across all methods is Stable Diffusion v2.1. For our method, in the stage of learning the personal subspace, we first fine-tune both the U-Net and text encoder for 1000 steps to learn identity information, using a learning rate of $1e-5$. Subsequently, we perform prompt tuning for 50 steps with a learning rate of $1e-3$. In the stage of optimizing universal privacy cloaks, we employ DDIM for the sampling process in the diffusion model, with a total of 50 sampling steps. For updating the universal cloak, we set the total training iterations to 200, with 10 inner iterations for gradient aggregation in each iteration, and the step size $\alpha$ is set to 0.05. We use the protected images with cloaks for personalized text-to-image generation, adopting DreamBooth as the default personalization technique. After fine-tuning for 1,000 steps, we generate images to measure the defense performance.

## C  DERIVATION OF THE OPTIMIZATION OBJECTIVE FOR THE ANCHOR POINTS

Here, we provide a detailed derivation from Eq. equation 4 to Eq. equation 5. Given a set of images $\{x_i\}_{i=1}^N$ and an identity condition $c_{ID}$, our goal is to learn a corresponding text conditions $c_i$ for each image $x_i$ such that $c_i$ best describes $x_i$. Formally, this can be expressed as:

$$c_i = \arg\max_c p(c|x_i, c_{ID}). \tag{18}$$

Directly optimizing the posterior probability $p(c|x_i, c_{ID})$ is intractable. Following Wan et al. (2024), using Bayes' theorem, we decompose it as:

$$p(c|x_i, c_{ID}) = \frac{p(x_i, c_{ID}|c) \cdot p(c)}{p(x_i, c_{ID})}. \tag{19}$$

Here, the denominator $p(x_i, c_{ID})$ acts as a normalization constant $Z$, since $x_i$ and $c_{ID}$ are conditionally independent given $c$. Thus, the posterior probability is proportional to:

$$p(c|x_i, c_{ID}) \propto p(x_i|c, c_{ID}) \cdot p(c). \tag{20}$$

We assume a uniform prior over $c$, i.e., $p(c) = \frac{1}{K}$. Under this assumption, the prior term becomes a constant, and the optimization objective simplifies to:

$$c_i = \arg\max_c p(x_i|c, c_{ID}). \tag{21}$$

Diffusion models maximize the likelihood indirectly by minimizing the noise prediction error. Let $\epsilon_\theta$ denote the denoising network, $\epsilon$ the true noise, $x_{i,t}$ the noisy version of $x_i$ at diffusion step $t$, and $\theta^*$ the frozen parameters from the identity learning stage. Following the diffusion training objective Ho et al. (2020), maximizing $p(x_i|c, c_{ID})$ is equivalent to minimizing:

$$\min_{c_i} \mathbb{E}_{\epsilon,t} \|\epsilon - \epsilon_{\theta^*}(x_{i,t}, t, c_i)\|_2^2. \tag{22}$$

Extending this to all images $\{x_i\}_{i=1}^N$, the final optimization objective becomes:

$$\min_{\{c_i\}_{i=1}^N} \mathbb{E}_{\epsilon,t} \sum_{i=1}^N \|\epsilon - \epsilon_{\theta^*}(x_{i,t}, t, c_i)\|_2^2 \tag{23}$$

Given that $c_{ID}$ serves as the point representing the individual's core identity information, which is typically expected to be highly correlated with the content of the image, we assume that $c_{ID}$ and $c_i$ are very close in the textual space. Therefore, the iterative solution for $c_i$ can be initialized from $c_{ID}$. This initialization provides a strong starting point for optimizing Eq. equation 5, ensuring faster convergence and better alignment with the image content.

## D  CLOAK UPDATING STRATEGY

Table 5: Ablation study on the gradient updating strategy.

| Method | BRISQUE↑ | ISM↓ | FDFR↑ | SER-FIQ↓ |
|---|---|---|---|---|
| w/o gradient aggregation | 26.62 | 0.378 | 0.109 | 0.554 |
| w/ gradient aggregation | 31.05 | 0.364 | 0.210 | 0.501 |

When updating the cloak, the Projected Gradient Descent (PGD) technique Madry (2017) is commonly employed, as follows:

$$\delta_i = \text{clip}_\delta^\eta(\delta_{i-1} + \alpha \cdot \text{sign}(\nabla_\delta \mathcal{L}(x_t, \delta))), \tag{24}$$

where the clip operation constrains the pixel values of $\delta$ within an $\eta$-ball around the original values, and $\mathcal{L}$ refers to the optimization objective defined in Eq. (12).

However, in our practice, directly applying PGD to optimize the cloak can lead to suboptimal results due to gradient instability. During each optimization iteration, a small batch of latents $x_t$ is sampled from Gaussian noise $x_T$. The significant variations between minibatches of $x_t$ sampled in different

---

**Algorithm 3** Optimizing Identity-specific Universal Cloak with Stochastic Gradient Aggregation

---

**Require:** Customized diffusion model $\theta^*$, individual subspace $Q(c)$, training iterations $N$, inner iterations for gradient aggregation $M$, noise budget $\eta$, PGD step size $\alpha$      ▷ identity-specific universal cloak $\delta$ for identity $k$

1: **Initialize:** $\delta = 0$
2: **for** $n = 1$ **to** $N$ **do**
3:      $\delta_1^{\text{inner}} \leftarrow \delta$
4:      $g^{\text{Aggs}} = 0$
5:      **for** $m = 1$ **to** $M$ **do**
6:          Sample $c \sim Q, t \in U(0, T)$
7:          Sample $x_t = \texttt{sample}(\theta^*, t, c)$
8:          Obtain $\hat{x}_t' = \texttt{applyCloak}(x_t, \delta_m^{\text{inner}})$
9:          Compute grad. $g_x = \nabla_{\hat{x}_t'} \|\epsilon_\theta^*(x_t, t, c) - \epsilon_\theta^*(\hat{x}_t', t, c)\|_2^2$
10:          $\delta_{m+1}^{\text{inner}} \leftarrow \text{clip}_\delta^\eta(\delta_m^{\text{inner}} + \alpha \cdot \text{sgn}(g_x))$
11:          $g^{\text{Aggs}} \leftarrow g^{\text{Aggs}} + g_x$
12:      **end for**
13:      $\delta \leftarrow \text{clip}_\delta^\eta(\delta + \alpha \cdot \text{sgn}(g^{\text{Aggs}}))$
14: **end for**
15: **Return:** Universal cloak $\delta^*$ for identity $k$

---

iterations cause instability in the gradients backpropagated through the loss function. Additionally, the use of the sign operation within the PGD framework introduces quantization errors, further exacerbating the instability caused by gradient fluctuations. As a result, the update directions become inconsistent, undermining the optimization process.

To address this issue, we adopt a strategy similar to Liu et al. (2023) and aggregate multiple small-batch gradients to update the cloak, thereby alleviating the aforementioned problem. Specifically, our strategy involves an inner-outer iteration framework. In the inner loop, we perform multiple optimization iterations, each involving sampling a minibatch of $x_t$ and computing the loss to obtain diversified gradients. Following Liu et al. (2023), we also introduce a pre-search step within each inner iteration, where a surrogate cloak $\delta^{\text{inner}}$ of $\delta$ is preliminarily updated. This step has been shown to enhance the generalization ability of universal cloaks. In each outer loop iteration, we aggregate the gradients collected from the inner loop to obtain a stable and reliable gradient, which is then used to update the universal cloak. By accumulating gradient information over multiple rounds, the aggregated gradient estimates become more accurate with reduced variance. This suppresses gradient instability and makes the optimization process of the universal cloak more effective. The complete optimization process with stochastic gradient aggregation is outlined in Algorithm. 3.

We also conduct an ablation study on the gradient updating strategy by replacing it with naive PGD. The results, as shown in Table 5, demonstrate that the gradient aggregation strategy has a significant impact on the effectiveness of the generated identity-specific cloak.

## E EXTENDED EXPERIMENTS

### E.1 GENERALIZABILITY TO STYLE IMITATION TASK

To test the generalizability of ID-Cloak to broader domains, we have conducted additional experiments on the style imitation protection task.

**Setup.** We performed this experiment on the WikiArt dataset following style protection works Wan et al. (2024); Liang et al. (2023). We randomly selected 40 artists with distinct styles. For each artist, we used 10 stylistically consistent artworks to train a style-specific cloak and a separate set of 10 artworks for evaluation. We use ten different test prompts related to the style of painting to evaluate the model's performance as Wan et al. (2024).

**Evaluation Metrics.** We use four metrics to comprehensively measure the effectiveness of style protection. First, visual quality was assessed using BRISQUE($\uparrow$) and the LAION aesthetic predictor($\downarrow$).

Second, stylistic consistency with the original works was measured with CLIP-I($\downarrow$). Finally, FID($\uparrow$) was used to evaluate distributional similarity.

**Results.** The results in Table 6 demonstrate that ID-Cloak is a generalizable framework applicable to protecting artistic styles, significantly broadening the impact of our work. Our method outperforms all comparison methods in disrupting style imitation, as evidenced by a much lower style similarity (CLIP-I) and higher Frechet Inception Distance (FID). Notably, compared to the strongest competitor (Anti-DB+), ID-Cloak increases the FID score by 27.9% and achieves a 2.7 times greater reduction in style similarity (CLIP-I). This substantial leap in performance confirms that our method can effectively insulate an artist's signature style from being learned and replicated by generative models.

Table 6: Style imitation protection performance on WikiArt dataset, averaged across 10 diverse prompts.

| Method | WikiArt | | | |
| --- | --- | --- | --- | --- |
| | **CLIP-I** ($\downarrow$) | **FID** ($\uparrow$) | **BRISQUE** ($\uparrow$) | **Aesthetic** ($\downarrow$) |
| Anti-DB | 63.35 | 92.62 | 23.90 | 5.66 |
| SimAC | 64.19 | 90.36 | 23.02 | 5.70 |
| MetaCloak | 63.91 | 91.71 | 22.15 | 5.71 |
| Anti-DB+ | 62.74 | 99.91 | 24.93 | 5.54 |
| SimAC+ | 63.28 | 99.60 | 25.15 | 5.55 |
| MetaCloak+ | 63.67 | 91.81 | 22.73 | 5.68 |
| **Ours** | **59.46** | **127.83** | **29.42** | **5.22** |
| Clean (Original) | 64.64 | — | 21.91 | 5.84 |

### E.2 ROBUSTNESS AGAINST COMMON IMAGE TRANSFORMATIONS

In real-world applications, an adversary may apply common image processing techniques to circumvent the protective cloak. To assess the resilience of our proposed method, we conduct evaluation of ID-Cloak's robustness against three simple image transformations: Gaussian blurring, JPEG Compression, and Smoothing with uniform noise.

The results in Table 7 demonstrate the superior robustness of our method. First, our method consistently demonstrates superior robustness compared to its competitors. While the effectiveness of all methods is expectedly reduced by these transformations, our approach significantly outperforms every competitor across all metrics. For instance, under Gaussian blurring, ID-Cloak maintains an FDFR of 0.086, approximately 2.7 times higher than the next best method. Second, our method provides effective protection after these transformations. Compared to the clean reference (ISM 0.525), our method's protection remains highly effective after transformation, with ISM scores of 0.435 (blurring), 0.444 (JPEG) and 0.426 (smoothing).

We attribute this enhanced resilience to our core approach: by perturbing the semantic identity subspace rather than optimizing for pixel-level artifacts, the protection becomes more deeply embedded with the core features of the identity, making it less susceptible to simple image transformations.

### E.3 QUANTITATIVE EVALUATION ON DIVERSE PROMPTS

To assess the robustness of our protection method against a wide range of potential generation contexts that attackers might utilize, we expanded our evaluation to a diverse set of 20 text prompts, curated from prior works Wan et al. (2024); Kong et al. (2024) and further extended to cover a broad spectrum of scenarios, such as simple portraits, artistic styles, and complex, context-rich scenarios. The full list of prompts used in this evaluation is provided in Table 9.

The quantitative results are presented in Table 8. Our method consistently outperforms all comparison methods, demonstrating that ID-Cloak's effectiveness is not limited to a few simple prompts but holds across a wide variety of potential attack prompts.

Table 7: Robustness to common image transformations on VGGFace2 dataset.

| Transformation | Method | ISM↓ | FDFR↑ | SER-FIQ↓ | BRISQUE↑ |
|---|---|---|---|---|---|
| Gaussian Blur | Anti-DB+ | 0.478 | 0.030 | 0.658 | 26.734 |
| | SimAC+ | 0.480 | 0.032 | 0.654 | 30.920 |
| | MetaCloak+ | 0.463 | 0.029 | 0.648 | 28.642 |
| | Ours | **0.435** | **0.086** | **0.608** | **31.434** |
| JPEG Compression | Anti-DB+ | 0.463 | 0.023 | 0.668 | 20.704 |
| | SimAC+ | 0.465 | 0.024 | 0.664 | 22.472 |
| | MetaCloak+ | 0.461 | 0.021 | 0.665 | 23.582 |
| | Ours | **0.444** | **0.033** | **0.638** | **25.999** |
| Smoothing | Anti-DB+ | 0.482 | 0.020 | 0.668 | 27.259 |
| | SimAC+ | 0.478 | 0.028 | 0.670 | 31.538 |
| | MetaCloak+ | 0.459 | 0.021 | 0.654 | 30.504 |
| | Ours | **0.426** | **0.077** | **0.599** | **31.580** |
| No Transformation | Ours | 0.364 | 0.210 | 0.501 | 31.051 |
| Clean Reference | / | 0.525 | 0.020 | 0.691 | 8.418 |

Table 8: Quantitative results aggregated over 20 diverse prompts on VGGFace2 dataset.

| **Method** | **ISM ($\downarrow$)** | **FDFR ($\uparrow$)** | **SER-FIQ ($\downarrow$)** | **BRISQUE ($\uparrow$)** |
|---|---|---|---|---|
| Anti-DB+ | 0.255 | 0.229 | 0.432 | 28.985 |
| SimAC+ | 0.261 | 0.244 | 0.432 | 30.161 |
| **Ours** | **0.200** | **0.363** | **0.320** | **33.874** |

### E.4 Ablation Study on Perturbation Budget

To investigate the impact of the perturbation budget on the performance of ID-Cloak, we conduct an ablation study by varying the noise perturbation scale $\eta$. This analysis helps to understand the trade-off between protection strength and the perceptual quality of the protected images. The experiments are performed on the VGGFace2 dataset. The results, presented in Table 10, reveal a clear and flexible trade-off. As the perturbation budget $\eta$ increases from 0 to $24/255$, the protection effectiveness steadily improves. Particularly, a significant increase in protection effectiveness was observed when the budget was increased from $12/255$ to $16/255$, where the FDFR score more than doubled from 0.098 to 0.210. Therefore, we selected $\eta = 16/255$ as our default setting for the main experiments, as it provides a compelling balance between robust identity protection and high perceptual quality.

## F COMPUTATIONAL COST ANALYSIS

To contextualize the practical applicability of our method, we provide a detailed analysis of its computational overhead. Table 11 presents a comparison of the runtime and GPU memory requirements for ID-Cloak against image-specific baselines. All performance metrics were benchmarked on a single NVIDIA H100 GPU.

In terms of execution time cost, image-specific methods have a linear time cost ($O(n)$), where the total protection time scales directly with the number of images ($n$). In contrast, ID-Cloak has a constant time cost ($O(1)$). Based on results, ID-Cloak becomes more time-efficient than SimAC after ~20 images and more efficient than Anti-DB after ~34 images, which means for users with a large or growing collection of photos, ID-Cloak is orders of magnitude more efficient. In terms of GPU memory cost, the GPU memory required during the cloak generation phase is comparable to image-specific methods, as the core operation involves a single backward pass through the U-Net.

| | Prompt |
|---|---|
| 1 | a photo of *sks* person. |
| 2 | dslr portrait of *sks* person. |
| 3 | an impressionistic depiction of *sks* person. |
| 4 | an abstract representation of *sks* person. |
| 5 | a cyberpunk style photo of *sks* person. |
| 6 | a realistic painting of *sks* person. |
| 7 | a concept art of *sks* person. |
| 8 | a headshot photo of *sks* person. |
| 9 | a caricature sketch of *sks* person. |
| 10 | a digital portrait of *sks* person. |
| 11 | *sks* person selfie standing under the pink blossoms of a cherry tree. |
| 12 | *sks* person in a chef's outfit, cooking in a kitchen. |
| 13 | *sks* person paddling a canoe on a tranquil lake. |
| 14 | *sks* person playing with their pet dog. |
| 15 | photo of *sks* person taking a shot in basketball. |
| 16 | *sks* person selfie with eiffel tower in the background. |
| 17 | *sks* person in an astronaut suit, floating in a spaceship. |
| 18 | *sks* person dressed in a firefighter's outfit, a raging forest fire in the background. |
| 19 | *sks* person wearing Victorian-era clothing, reading a book in a classic British library. |
| 20 | *sks* person dressed as a knight, standing in a medieval castle. |

Table 9: Full test prompt list for evaluation on more diverse set of text prompts.

Table 10: Ablation study on the perturbation budget $\eta$ on the VGGFace2 dataset. The default setting used in our main experiments is marked with an asterisk (*).

| Noise Budget ($\eta$) | ISM ($\downarrow$) | FDFR ($\uparrow$) | SER-FIQ ($\downarrow$) | BRISQUE ($\uparrow$) |
|---|---|---|---|---|
| 0 | 0.525 | 0.020 | 0.691 | 8.418 |
| 4/255 | 0.483 | 0.026 | 0.664 | 16.451 |
| 8/255 | 0.437 | 0.050 | 0.621 | 20.177 |
| 12/255 | 0.403 | 0.098 | 0.554 | 25.043 |
| 16/255* | 0.364 | 0.210 | 0.501 | 31.051 |
| 24/255 | 0.285 | 0.460 | 0.317 | 35.675 |

Table 11: Computational cost comparison between image-specifc protection and identity-specific protection. (Time in minutes)

| Method | Initial Cloak Crafting Time | Per-Image Protection Time | Total Time (for $n$ images) | Total Time ($n = 30$) | Total Time ($n = 100$) | GPU Memory (Peak) |
|---|---|---|---|---|---|---|
| Anti-DB | — | ~1.5 mins | ~ $1.5n$ mins | ~45 mins | ~150 mins | ~20 GB |
| SimAC | — | ~2.5 mins | ~ $2.5n$ mins | ~75 mins | ~250 mins | ~27 GB |
| **ID-Cloak** | ~50 mins (5 + 45) | 0 (Instantaneous) | ~ 50 mins | ~50 mins | ~50 mins | ~23 GB |

## G  ADDITIONAL QUALITATIVE COMPARISONS

Additional comparison results from our main experiment are presented in Figure 3 and 4 for the VGGFace2 dataset, and in Figure 5 and 6 for the CelebA-HQ dataset. These results, in conjunction with those presented in Figure 2 of the main paper, substantiate the effectiveness of our method in generating identity-specific cloaks that robustly protect facial privacy and demonstrate its strong generalization capability across all face images of an individual.

To further demonstrate that our method imposes robust and consistent protections against a wide range of potential prompts that attackers might use, we have also shown some qualitative comparison results on some more complex prompts in Figure 7, 8, and Figure 9.

## H   LIMITATIONS AND FUTURE WORK

While our proposed ID-Cloak method demonstrates significant advancements in achieving identity-specific privacy protection, certain limitations warrant further consideration and future research. For example, the protection effectiveness may decrease when the images to be protected differ significantly (e.g., facial proportion within the image, background scene, etc.) from the few-shot input images used to model the identity subspace and generate the universal cloak. This is because our method models the identity subspace based on these limited input images, and thus the fitted image distribution is primarily constructed around them. Any image that deviates considerably from the input images might fall outside our fitted distribution, leading to a reduction in protection efficacy. Future research could explore more robust identity subspace modeling techniques to better generalize to a wider range of image variations.

**Generalization to Tuning-Free Personalization Methods.** Recent advancements in personalization have seen the rise of popular tuning-free approaches, such as IP-Adapter Ye et al. (2023), PhotoMaker Li et al. (2024b), PuLID Guo et al. (2024), etc. To investigate whether ID-Cloak can effectively generalize to these methods, we evaluated its protective performance against the widely-used IP-Adapter, a representative tuning-free personalization model. We performed a direct comparison, generating images with IP-Adapter conditioned on (a) original, unprotected images and (b) their ID-Cloak protected counterparts.

As shown in Table 12, the application of ID-Cloak leads to a substantial degradation in personalization quality. The marked decrease in ISM and SER-FIQ scores confirms that our cloak successfully disrupts identity replication even for a powerful tuning-free model like IP-Adapter. This result demonstrates that the protective mechanism of ID-Cloak generalizes beyond tuning-based methods.

However, It is crucial to recognize the fundamental differences between tuning-based and tuning-free personalization methods. Our paper's primary focus is on providing a comprehensive defense for the tuning-based family (DreamBooth, LoRA, etc.), where we have demonstrated robust and consistent success. While our positive results against the tuning-free IP-Adapter highlight ID-Cloak's generalizability, the significant architectural differences between these two families mean that a dedicated defense for tuning-free models constitutes a separate research problem. We believe this is an important direction for future investigation.

Table 12: Protection Effectiveness on IP-Adapter (tuning-Free) on VGGFace2 dataset.

| Method | ISM $\downarrow$ | SER-FIQ $\downarrow$ |
|---|---|---|
| No protection | 0.291 | 0.632 |
| w/ ID-Cloak | **0.222** | **0.580** |

## I   THE USE OF LARGE LANGUAGE MODELS (LLMs)

In the preparation of this manuscript, large language models (LLMs) were used solely for writing assistance, including grammar checking, phrasing refinement, and formatting. The LLM was not involved in the conception, design, implementation, or analysis of the core methodology presented in this work. All content is original and authored by the listed authors.

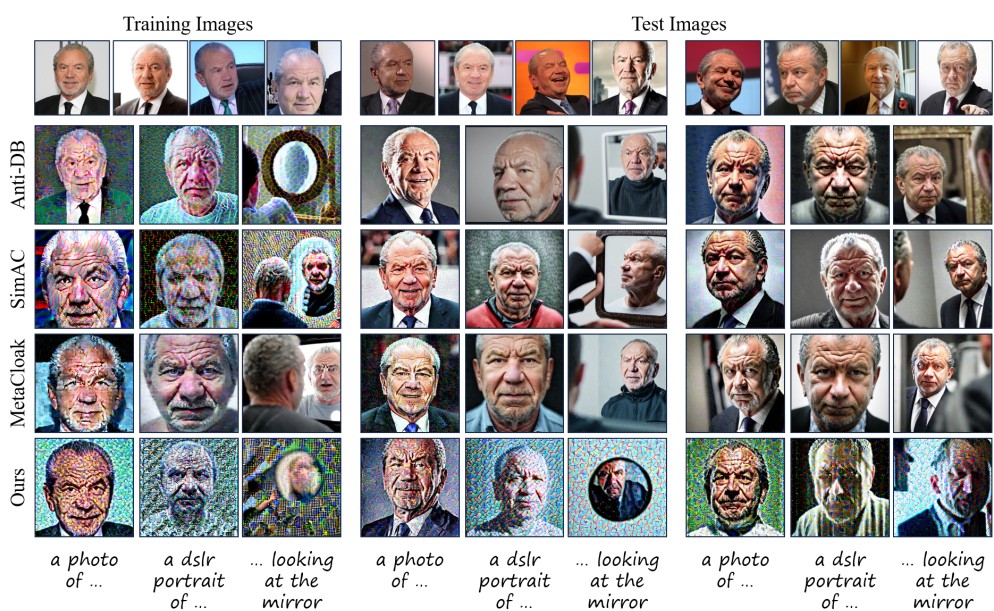

Figure 3: Additional qualitative results on VGGFace2 dataset. The cloaks are generated from the images of training set, then applied on the same training set and different test sets respectively. Each row represents a method, and each column represents a different test prompt.

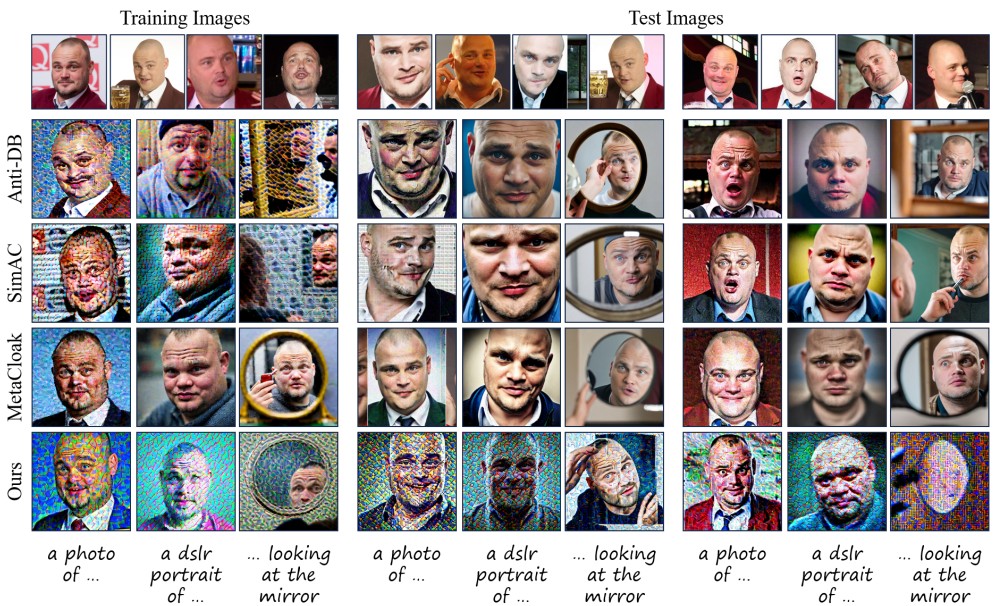

Figure 4: Additional qualitative results on VGGFace2 dataset. The cloaks are generated from the images of training set, then applied on the same training set and different test sets respectively. Each row represents a method, and each column represents a different test prompt.

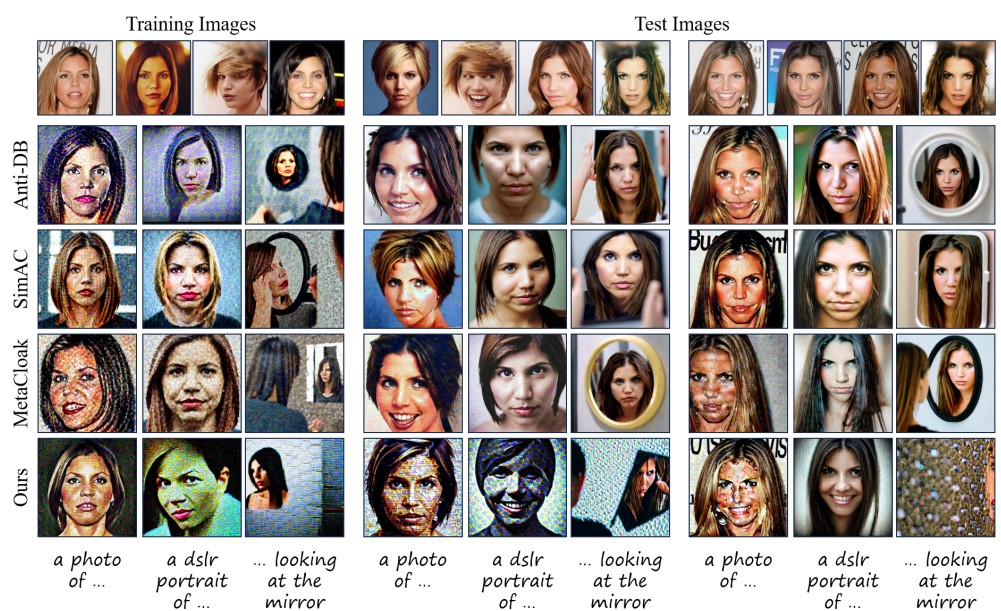

Figure 5: Additional qualitative results on CelebA-HQ dataset. The cloaks are generated from the images of training set, then applied on the same training set and different test sets respectively. Each row represents a method, and each column represents a different test prompt.

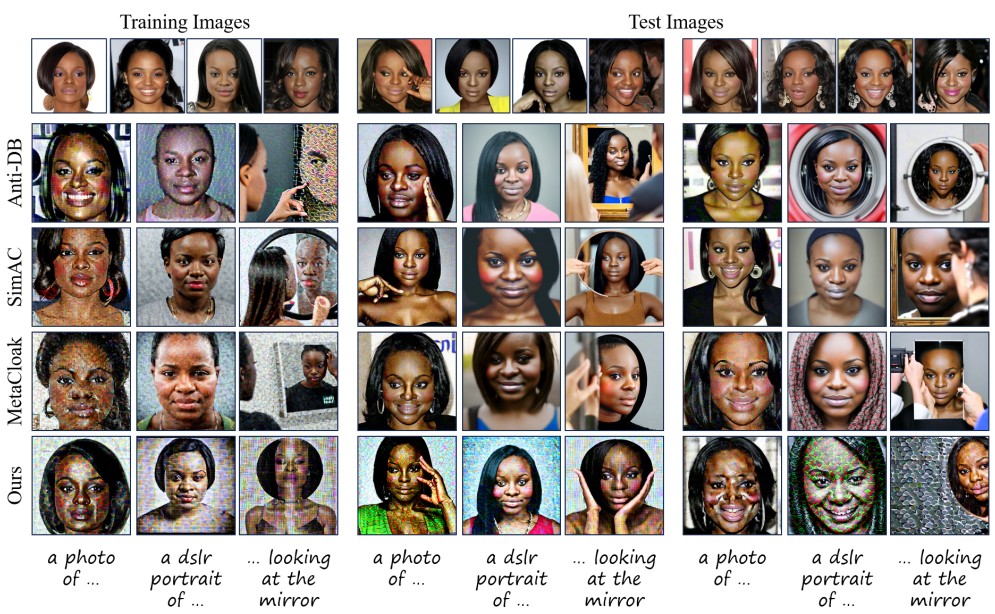

Figure 6: Additional qualitative results on CelebA-HQ dataset. The cloaks are generated from the images of training set, then applied on the same training set and different test sets respectively. Each row represents a method, and each column represents a different test prompt.

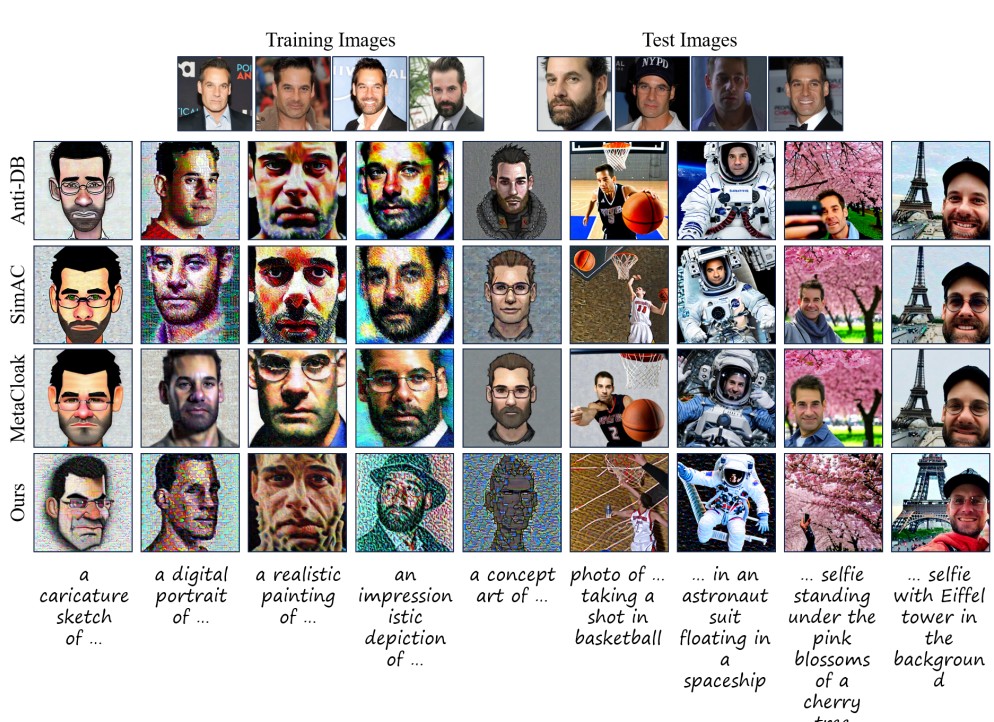

Figure 7: Additional qualitative results on more complex attack prompts.

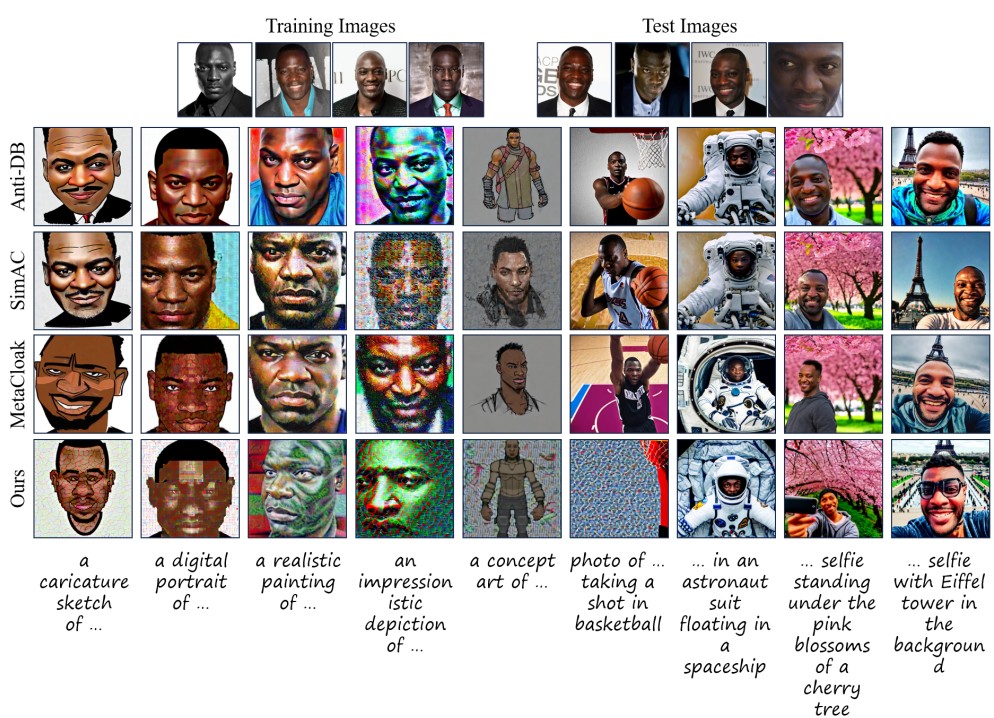

Figure 8: Additional qualitative results on more complex attack prompts.

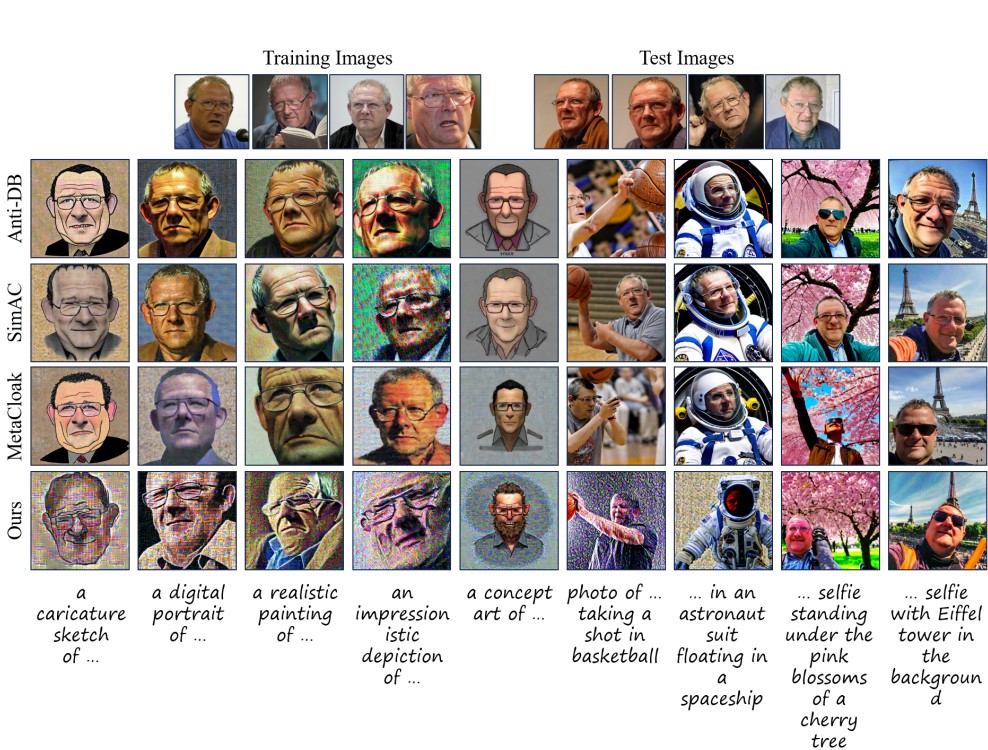

Figure 9: Additional qualitative results on more complex attack prompts.

