# OpenReview forum: "ID-Cloak: Crafting Identity-Specific Cloaks Against Personalized Text-to-Image Generation"
_ICLR.cc/2026/Conference — ICLR 2026 Conference Withdrawn Submission_

### Official Review · Reviewer_RMKR · 2025-10-27

**Soundness:** 3
**Presentation:** 3
**Contribution:** 3
**Rating:** 6
**Confidence:** 4

**Summary:**

In this paper, the authors propose an anti-personalization approach that creates identity-specific cloaks (ID-Cloak) designed to safeguard all images belonging to a particular identity. The proposed approach consists of two stept. First, they model an identity subspace that preserves personal commonalities and learns diverse contexts to capture the image distribution to be protected. Second, they craft identity-specific cloaks with the proposed novel objective that encourages the cloak to guide the model away from its normal output within the subspace. Extensive experiments (both qualitative and quantitative) show that their approach achieves robust protection across all images of an identity using a single universal mask.

**Strengths:**

Here are the paper's strength:
- the paper is well-written and well-documented
- the proposed approach is novel and scientifically sound (a novel protection paradigm)
- the review of the state of the art covers most of the relevant literature in the field
- the experimental evaluation is extended and confirms the starting hypothesis

**Weaknesses:**

Here are the paper's weaknesses:
- While the empirical results are convincing, the authors should provide more formal justification why the learned subspace generalizes effectively across unseen identity sample
- The experimental validation is currently limited to face datasets. To establish the proposed approach's generalizability, it is essential to demonstrate its effectiveness on other personalized domains (e.g. objects, artwork).
- Some aspects require an improved clarity and a more detailed explanation, like the following statement in section 3.2.1 (lines 195-197): "This formulation shifts the focus from complex image distributions to a more structured and interpretable text-based representation. Specifically, it allows us to model the protected identity’s image distribution by focusing on a semantically meaningful subspace in the text embedding space."

**Questions:**

Besides the weaknesses mentioned above, here is the list of my other concerns:
- Eq. 7: The left hand side of the equation should be $\delta^*$, since it is the result of an optimization process. Furthermore, it should be made consistent with eq. 1
- In order to avoid confusion, in algorithm 2, please change the letter N (training iterations) by other letter, since you used N before to refer to the number of anchor points.
- You claim you model a Gaussion distribution from 4 anchor points. Is this enough?
- How sensitive is the protection performance to the number of anchor points or the dimensionality of the modeled identity subspace?
- Can you provide more intuition or analysis on why the Gaussian identity subspace effectively captures both core identity and contextual diversity?
- Could you show visualizations or quantitative metrics (e.g., PSNR, LPIPS) demonstrating that the added cloak remains imperceptible to human observers while still being effective against personalization attacks?
- Have you identified any situations where the method fails to provide sufficient protection (e.g., unusual lighting, occlusion, or profile faces)? What might cause these failures?
- When adapting image-specific baselines into their “universal” variants, how do you ensure a fair comparison with ID-CLOAK?

---

### Official Review · Reviewer_JiPu · 2025-10-31

**Soundness:** 3
**Presentation:** 3
**Contribution:** 3
**Rating:** 6
**Confidence:** 4

**Summary:**

The paper proposes ID-Cloak, a practical privacy defense for personalized text-to-image models. Instead of crafting image-specific perturbations, it learns an identity-specific universal cloak from only a few images. The key idea is to model an identity subspace in the text embedding space (a Gaussian built from prompt-tuned anchors) to approximate all images of the person, and then optimize a universal perturbation that forces the diffusion model to deviate from its normal outputs within this subspace. Experiments on CelebA-HQ and VGGFace2 show stronger and more transferable protection than adapted image-specific baselines.

**Strengths:**

1. The paper tackles a very realistic and high-impact problem: moving from “protecting a single posted image” to “protecting an identity across many current and future images.” This identity-level setting is much closer to how photos actually appear and spread on social platforms, and it addresses the core scalability gap in prior image-specific cloaking methods.
2. The shift from doing adversarial perturbation directly in image space to first constructing an identity subspace in the text/semantic embedding space is conceptually elegant. By operating over the semantic region corresponding to “this person,” the method aims to protect not just the few available photos but the whole semantic neighborhood that personalized T2I models would try to learn.
3. The experimental section is reasonably comprehensive: it compares against adapted image-specific baselines, evaluates different prompts, includes cross-model / cross-personalization transfer, and shows that the proposed identity-aware formulation gives more stable protection than prior work.

**Weaknesses:**

1. The core assumption “identity subspace is equal to a Gaussian estimated from 4 images” feels under-justified. With such a small sample, the distribution is very sparse, and it is unclear whether it can really cover hard real-world variants of the same person (extreme pose, strong makeup, occlusion, unusual lighting, profile views). The paper should either validate the coverage or relax the parametric assumption.
2. The evaluation mostly measures “did the generated face become worse / less identifiable,” but it does not report imperceptibility / usability on the original shared image. In practice, users want to post aesthetically normal photos, not visibly perturbed ones. Without reporting distortion budgets, perceptual scores, or even human judgments on the cloaked inputs, the deployability of the method is hard to assess.
3. The generalization is still narrow: results are shown on SD 1.5 / 2.1 and a small set of personalization pipelines. To claim identity-level protection in the wild, it would be important to test on a more recent, higher-capacity model such as SDXL and SDv3 to show the approach is not tied to one latent-diffusion family or certain text encoder(s).

**Questions:**

N/A

---

### Official Review · Reviewer_vz9Z · 2025-11-01

**Soundness:** 2
**Presentation:** 2
**Contribution:** 2
**Rating:** 4
**Confidence:** 3

**Summary:**

The paper proposes ID-Cloak, a method for identity-specific privacy protection against malicious personalized text-to-image (T2I) generation. Unlike prior image-specific defenses that require crafting a unique perturbation per image, ID-Cloak learns a single universal cloak per identity from a few reference images. The core idea involves modeling an identity subspace in the text embedding space using a Gaussian distribution over anchor embeddings derived from the reference images. A universal perturbation is then optimized to maximize divergence between the model’s output on clean vs. cloaked images within this subspace. Experiments show strong performance across datasets (VGGFace2, CelebA-HQ), personalization methods (DreamBooth, LoRA, Textual Inversion), and cross-model transfer.

**Strengths:**

1. The shift from image-specific to identity-specific protection is well-motivated and addresses a critical scalability bottleneck in real-world deployment, where users may have hundreds of public images.

2. The use of a learned identity subspace in the text embedding space is a smart way to implicitly model the distribution of all possible images of a person without requiring explicit data augmentation.

3. The paper includes comprehensive experiments: comparison with strong baselines extended to universal variants. Robustness to prompt variation, model transfer, and personalization techniques. Ablation studies validating design choices.

4. The O(1) cost for protecting n images is a significant practical advantage over O(n) image-specific methods.

**Weaknesses:**

1. While the paper claims to be the “first” to investigate identity-specific cloaks, [1] already proposed a person-specific universal perturbation for face recognition protection. Similarly, Universal Adversarial Perturbations (UAPs) have been studied extensively in classification and face recognition [2,3]. The paper’s core idea—learning a single perturbation per class/identity—is not fundamentally new, though its application to personalized T2I misuse is timely.

2. The identity subspace is modeled as a simple Gaussian over prompt-tuned embeddings—a reasonable but not highly innovative choice. The optimization objective (maximizing noise prediction discrepancy) is a standard adversarial loss adapted to diffusion models, similar in spirit to Anti-DreamBooth and MetaCloak.

3. Experiments are limited to face images; generalization to objects or styles (beyond a small WikiArt experiment) is unclear.

4. The “identity subspace” is learned using the same model (SD v2.1) used for attack evaluation, which may overstate effectiveness in black-box settings.

5. Human perceptual studies are absent; BRISQUE/SER-FIQ are imperfect proxies for visual quality.

[1] OPOM: Customized Invisible Cloak towards Face Privacy Protection
[2] Universal Adversarial Perturbations
[3] Enhancing Generalization of UAP through Gradient Aggregation

**Questions:**

1. How does ID-Cloak fundamentally differ from OPOM, which also learns a single identity-specific perturbation for privacy? The related work section mentions OPOM only in passing under UAPs but does not engage with its identity-specific formulation.

2. The identity subspace assumes that all images of a person can be modeled by a Gaussian in text embedding space. How sensitive is performance to this assumption? Have you tried more flexible distributions (e.g., mixture models, normalizing flows)?

3. The proposed method requires fine-tuning the full model (U-Net + text encoder) in the identity learning stage (Sec. B). This is computationally heavy (~50 mins). Can this be avoided (e.g., using only Textual Inversion)? why the Custom Diffusion or similar personalization methods are not used? they are pretty fast methods.

4. For tuning-free personalization methods like IP-Adapter or PhotoMaker, your results (Table 12) show only modest degradation. Do you believe a different defense strategy is needed for these architectures?

5. Why you did not tried newer SD or DiT based methods?

---

### Official Review · Reviewer_gcn4 · 2025-11-06

**Soundness:** 2
**Presentation:** 3
**Contribution:** 2
**Rating:** 4
**Confidence:** 4

**Summary:**

This paper addresses privacy risks posed by personalized text-to-image (T2I) models (e.g., DreamBooth), where malicious users can generate fake images from personal photos. The paper proposes ID-Cloak—the first identity-specific universal cloak that protects all images of a single identity using only a few (e.g., 4) reference images. Its workflow has two key steps: Identity Subspace Modeling and Cloak Optimization. Experiments approve the effectiveness of the proposed method.

**Strengths:**

1. Paradigm Shift from Image-Specific to Identity-Specific Protection: It pioneers the transition from inefficient "image-level" cloaks (one per image, as in prior works like Anti-DreamBooth) to "identity-level" universal cloaks.
2. Data Efficiency with Minimal Input Requirement: ID-Cloak only needs a small set of reference images to generate a cloak that protects all images of the target identity.
3. This paper is well-written.

**Weaknesses:**

1. Insufficient Targeting for Tuning-Free Personalization Methods: While ID-Cloak shows some defensive effect against tuning-free methods, its core design focuses on tuning-based techniques.
2. Trade-off Dilemma in Perturbation Budget: The method relies on a perturbation budget (η) that requires balancing protection strength and visual imperceptibility.
3. The experiments only under standard conditions. There is no testing on extreme scenarios, such as low-quality reference images (blurred, occluded, or low-light).

**Questions:**

1. Insufficient Targeting for Tuning-Free Personalization Methods: While ID-Cloak shows some defensive effect against tuning-free methods, its core design focuses on tuning-based techniques.
2. Trade-off Dilemma in Perturbation Budget: The method relies on a perturbation budget (η) that requires balancing protection strength and visual imperceptibility.
3. The experiments only under standard conditions. There is no testing on extreme scenarios, such as low-quality reference images (blurred, occluded, or low-light).

---

### Note · Authors · 2025-12-09

I have read and agree with the venue's withdrawal policy on behalf of myself and my co-authors.